

# Cold wintertime air masses over Europe: Where do they come from and how do they form?

Tiina Nygård[1], Lukas Papritz[2], Tuomas Naakka[1,3,] Timo Vihma[1]

[1]Finnish Meteorological Institute, Helsinki, Finland
[2]Institute for Atmospheric and Climate Science, ETH Zurich, Zurich, Switzerland
[3]Department of Meteorology, Stockholm University, Stockholm, Sweden

*Correspondence to*: Tiina Nygård (tiina.nygard@fmi.fi)

**Abstract.** Despite the general warming trend, wintertime cold air outbreaks in Europe have remained nearly as extreme and as common as decades ago. In this study, we identify six principal 850 hPa cold anomaly types over Europe in 1979–2020
using self-organizing maps (SOMs). Based on extensive analysis of atmospheric large-scale circulation patterns combined with nearly two million kinematic backward trajectories, we show the origins and contributions of various physical processes to the formation of cold wintertime 850-hPa air masses. The location of the cold anomaly region is closely tied to the location of blocking; if the block is located farther in the east, also the cold anomaly is displaced eastwards. Considering air-mass evolution along the trajectories, the air parcels are typically initially (5–10 d before) colder than at their arrival in Europe, but
also initially warmer air parcels sometimes lead to cold anomalies over Europe. Most commonly the effect of adiabatic warming on the temperature anomalies is overcompensated by advection from regions that are climatologically colder than the target region, supported by diabatic cooling along the pathway. However, there are regional differences: cold anomalies over western Europe and southeastern Europe are dominantly caused by advection, and over eastern Europe by both advective and diabatic processes. The decadal-scale warming in the site of air mass origin has been partly compensated by enhanced
diabatic (radiative) cooling along the pathway to Europe. There have also been decadal changes in large-scale circulation patterns and air mass origin. Our results suggest that understanding future changes in cold extremes will require in-depth analyses on both large-scale circulation and the physical (adiabatic and diabatic) processes.

## 1 Introduction

Europe is vulnerable to cold spells in winter impacting many human activities, such as agriculture, transport as well as demand and supply of energy, which are tightly connected to economy. Cold conditions also directly affect human well-being and increase mortality rates more than summertime heat waves (Gasparrini et al., 2015). Despite the general warming trend, cold air outbreaks in Europe have remained as extreme and nearly as common as decades ago (Smith and Sheridan, 2020).

It has been recognized for relatively long that cold spells and outbreaks in Europe are associated with the North Atlantic
Oscillation (NAO) and Arctic Oscillation (AO). These indices represent the intensity of the eastward airflow from the northern North Atlantic (Walsh et al., 2001; Vavrus et al., 2006; Vihma et al., 2020), and the negative phases of NAO and AO are



known to be associated with westward migrating cold air masses of Siberian origin and enhanced meridional transport (Walsh et al., 2001; Vavrus et al., 2006). More recently, several studies have focused on the connections of cold spells and outbreaks with atmospheric blocking, indicating that winter cold spells in Europe are associated with blocking over the northern North

Atlantic, continental Europe or Ural region (Sillmann et al., 2011; Pfahl, 2014; Brunner et al., 2018; Vihma et al., 2020; Kautz et al., 2022; Sui et al., 2022). Blocking systems are quasi-stationary flow patterns with a large meridional flow component, and wintertime cold anomalies typically occur downstream or south of a blocking system (Kautz et al., 2022). The horizontal advection of cold air on the eastern flank of a block typically originates from higher latitudes (i.e., Arctic) or from cold, continental areas (mostly Russia) (Bieli et al., 2015), and the associated cold air masses can range up to several thousand

kilometers in scale (Walsh et al., 2001). In fact, impacts of NAO and blocking are difficult to separate, as stated by Kautz et al. (2022); blocking over the northern North Atlantic is highly correlated with the negative phase of the NAO, and the negative phase can be defined via a blocking pattern.

Compared to impacts of large-scale circulation, the physical (adiabatic and diabatic) processes contributing to the cold extremes in Europe in winter have so far been less studied. Based on trajectory analyses linked to European cold extremes,

Bieli et al. (2015) demonstrated that the air temperature along the trajectories is lowest in the source region, but the extremeness of the air mass with respect to its local surroundings increases when it enters regions with a milder climate. Along the pathway, horizontal and vertical transport, subsidence-induced warming and diabatic processes modify the air mass and may either amplify or dampen its extremeness (Bieli et al., 2015; Papritz, 2020; Vihma et al., 2020).

Definitions of cold extremes and cold air outbreaks vary across the literature. A good overview of the definitions previously

used is given in Smith and Sheridan (2020). In brief, 2 m temperature has typically been used to identify cold air outbreaks and extremes over land (Wheeler et al., 2011; Bieli et al., 2015; Ayarzagüena and Screen, 2016; Smith and Sheridan, 2020; Vihma et al., 2020), whereas the difference between 850 hPa/700 hPa and surface potential temperature has been used for marine cold air outbreaks (Kolstad and Bracegirdle, 2008; Papritz et al., 2015; Afargan-Gerstman et al., 2020). In addition, the meridional flux of cold polar air mass, with a threshold potential temperature of 280 K, has been analyzed in isentropic

coordinates (Kanno et al., 2016). Most previous studies have defined some criteria for the minimum geographical extent and duration of cold spells and outbreaks (Vavrus et al., 2006; Wheeler et al., 2011; Bieli et al., 2015; Ayarzagüena and Screen, 2016; Smith and Sheridan, 2020).

In this study, we analyze the relationship of 850 hPa cold anomalies with the large-scale circulation in winter and assess contributions of various physical processes to the formation of those cold air masses. Although the 850 hPa temperature

considered here does not have as direct impacts on human life as the 2 m temperature addressed in many previous studies, it represents large-scale air masses almost free of effects of the diurnal cycle and boundary layer processes. Specifically, we (i) identify the strongest cold anomalies in 850-hPa temperature over Europe during November–March 1979–2020, (ii) analyze





the large-scale circulation patterns as well as air mass origins and pathways behind those anomalies, (iii) reveal the contributions of different processes (i.e., horizontal and vertical transport, subsidence-induced warming and diabatic processes)

linked to the cold anomalies, and (iv) assess long-term trends in the occurrence of cold anomalies and the processes causing them. Thereby, we will especially focus on cold anomalies of an Arctic origin. We apply Self-Organizing Maps (SOM) to identify and cluster the cold air masses over Europe without predefined geographical subregions, thus providing a more flexible and objective perspective than previous studies. As a novel aspect, we quantify contributions of individual circulation patterns to the cold anomalies. These circulation patterns are not restricted to circulation indices, such as NAO and AO, or detection of

any single dynamical feature, such as blocking. Furthermore, we build a connection between circulation (streamlines), trajectories, and physical processes linked to cold air masses.

## 2 Data and methods

### 2.1 Data

Our analyses are based on the global atmospheric reanalysis ERA5 (C3S, 2017; Hersbach et al., 2020), provided by the

Copernicus Climate Change Service. In ERA5, a four-dimensional variational data assimilation method is applied to assimilate a variety of atmospheric observations into the Integrated Forecasting System (IFS) of the European Centre for Medium-Range Weather Forecasts (ECMWF). The spectral model resolution is T639, and there are 137 model levels in the vertical; we used data at 0.25° x 0.25° horizontal resolution for all analyses except for trajectory calculations, for which data at 0.5° x 0.5° horizontal resolution was used.


Cold anomalies were calculated from daily averages of 850 hPa temperature fields during extended winter, November–March, in 1979–2020. Anomalies were obtained by removing the climatological seasonal cycle, calculated by taking ±5 days running mean of 850 hPa temperature for each day in November–March and averaging this cycle over the period 1979–2020; this was done for each grid point separately. The climatology was smooth and neither strongly affected by individual cases nor having

abrupt steps.

Uncertainties of 850 hPa temperature in the operational IFS have been evaluated by Naakka et al. (2019), who found that the root-mean-square difference between 850 hPa temperature in IFS analyses and radiosonde observations was approximately 0.5°C over northern Europe. Hence, it is fair to suppose that uncertainty of 850 hPa temperature in ERA5, based on the IFS

model, is adequately small for the purpose of this study. In the analyses of physical processes behind the cold extremes, we utilized temperature $T$ and potential temperature $\Theta$ of ERA5 on the model levels.



Large-scale circulation was analyzed based on daily averages of mean sea level pressure and 500 hPa geopotential fields of ERA5. These variables have only a minor seasonal cycle during November–March, and therefore their seasonal cycle was not removed (except in the calculations of the Ural high (UH), described in the Supplementary material). Uncertainties of pressure and geopotential fields of ERA5 are assumed to be small, as in global atmospheric reanalyses in general.

## 2.2 Methods

### 2.2.1 SOM clustering

The identification and clustering of 850 hPa temperature anomaly and mean sea level pressure (MSLP) fields in the European domain shown in Fig. 1 were accomplished using the SOM method, developed by Kohonen (2001). SOM is an unsupervised learning method, thus a machine-learning approach, to determine generalized patterns in data. In meteorological applications, the SOM method provides physically meaningful composites of field patterns, preserving the probability density of the input data. We made separate SOM analyses for 850 hPa temperature anomaly and MSLP, results of which were later linked together, as demonstrated in Supplementary Fig. 1.

First, the input (850 hPa temperature or MSLP) data were re-gridded to an equal-area grid. Then an initial SOM array was created, containing nodes with random reference vectors of an equal dimension as the input data. Each input data vector was compared with these reference vectors. The reference vectors most similar to the input data vector were adjusted towards the input data vector until the reference vectors converged. As an outcome, an organized SOM array was created, with the most similar nodes (i.e., field patterns) situated close to each other and the most dissimilar ones in the corners. In this study, we analyze and show composites of the population of the input fields associated with the most similar node in the SOM array, not the output reference vectors of the SOM analysis. A more detailed description of the SOM method can be found in Kohonen (2001), Hewitson and Crane (2002) and Gibson et al. (2017).

Although the SOM technique provides an objective method for clustering, the choice of the SOM array size is always somewhat subjective (Alexander et al., 2010). With our choice of a 7 x 8 array, characteristic 850 hPa temperature anomaly patterns within Europe were captured with details (Fig. 2). With this SOM approach, we were able to cluster 850 hPa temperature anomalies according to their geographical location, extent, and shape, without any predefined locations of the anomalies. Having as many as 56 nodes for temperature anomalies enabled us to control which anomaly types (i.e., nodes) can be meaningfully merged in further analyses.

A SOM analysis based on MSLP was made to identify the dynamical conditions during and preceding the cold anomalies. Our comparison with different array sizes (7 x 8 and 5 x 6) indicated that an array smaller than that for 850 hPa temperature anomaly was justified, because MSLP fields contain less smaller-scale variability. Hence, a 5 x 6 SOM array with 30 characteristic MSLP patterns is used in this study (Supplementary Fig. 3). Composites of geopotential at 500 hPa were



calculated for all the nodes in the MSLP SOM array; the node numbers of MSLP (Supplementary Fig. 3) and 500 hPa geopotential SOM arrays (Supplementary Fig. 4) correspond to each other. Hereon, we refer to the SOM based on MSLP and the corresponding 500 hPa geopotential composites as circulation SOMs.

### 2.2.2 Selecting cold air masses and associating them with large-scale circulation

We selected the coldest anomalies in the 850 hPa temperature SOM array (Fig. 2) for further analyses using the 10[th] percentiles (Fig. 1) as regionally-varying thresholds. The percentiles were calculated using all the November–March time steps over the study period 1979–2020. Those 14 SOM nodes (marked in Fig. 2), with the composite 850 hPa temperature anomaly colder than the percentile threshold (Fig. 1) somewhere over continental Europe or the British Isles, were selected. These 14 cold anomalies were further amalgamated into six groups (Fig. 2) based on similarities in the location of the region with the anomaly being colder than the threshold. Hereon, we refer to those six groups as cold anomaly types.

We associated the circulation SOMs with the cold anomaly types using time information. We also made traditional, direct composites of MSLP and 500 hPa geopotential fields in each of the six cold anomaly types to be able to compare them to the individual SOM circulation types. This was done to investigate whether the direct MSLP and 500 hPa geopotential composites suffer from averaging of very different circulation types, and thus display an average circulation field, features of which do not occur as such. Data and methods linked to climate indices (NAO, AO, Greenland Blocking Index (GBI) and Ural high (UH)) are described in the Supplementary material.

### 2.2.3 Trajectory analyses

To analyze the origin, pathway and thermodynamic evolution of air masses contributing to the six cold anomaly types, we computed 10-day kinematic backward trajectories with the aid of the Lagrangian Analysis Tool (LAGRANTO; Sprenger and Wernli, 2015). Trajectories were initialized on the 850 hPa level at 12 UTC on the day when the cold anomaly occurred. Initialization points were distributed equidistantly in the horizontal with 50 km spacing within the cold anomaly, defined separately for each case as the area within the cold anomaly region where the 850 hPa daily mean temperature anomaly was lower than the 10[th] percentile (Fig. 1). Trajectories were then computed over a period of 10 days backward in time (spanning from 0 h to -240 h) using 3-hourly three-dimensional fields from ERA5. Note that only cold anomalies with at least 10 cold anomaly grid points on the 0.25° x 0.25° lat-lon grid were considered in the trajectory analyses. The number of cases for which trajectories were computed and the number of trajectories are shown in Table 1. Moreover, an example of an ensemble of trajectories is shown in Supplementary Fig. 6 for a selected case, illustrating the complexity of air parcel trajectories associated with one single event. In particular, it shows that air parcels of a cold anomaly region can originate from several distinct and distant geographical regions and undergo very different evolutions along their path.



For the further analyses, we interpolated temperature $T$, potential temperature $\Theta$ and climatological temperature $T_C$ to the trajectory positions. The model-level temperature climatology $T_C$ was computed separately for each synoptic time (e.g., 00 UTC, 03 UTC, etc.) as the ±5 day running mean including only timesteps with the given synoptic time and averaged over all years in the study period. This leads to a smoothly varying seasonal cycle while at the same time still maintaining the diurnal cycle.

To study how cold anomalies form along trajectories, we employed the framework by Röthlisberger and Papritz (2023a) and Röthlisberger and Papritz (2023b)**.** This framework allows quantifying the contributions of (i) advection from a climatologically warmer or colder region (hereafter *advection*), (ii) vertical motion (adiabatic expansion or compression; hereafter *adiabatic*), and (iii) diabatic processes (hereafter *diabatic*) to the emergence of temperature anomalies along kinematic trajectories. It is based on the thermodynamic energy equation formulated in terms of temperature anomalies and integrated in time along kinematic trajectories from the last time when the temperature anomaly was less than or equal to 0 K, thereafter referred to as time of anomaly genesis $t_g$. Specifically, the decomposition of the temperature anomaly reads

$$T'(x,t) = - \int_{t_g}^{t} \boldsymbol{v} \cdot \nabla_h T_C \, d\tau + \int_{t_g}^{t} \left[ \frac{\kappa T}{p} - \frac{\partial T_C}{\partial p} \right] \omega \, d\tau + \int_{t_g}^{t} \left( \frac{p}{p_0} \right)^{\kappa} \frac{D\theta}{Dt} \, d\tau + \text{res} \tag{1}$$

where $T'(x, t)$ is the temperature anomaly along the trajectory at time $t$, $\boldsymbol{v}$ and $\omega$ are the horizontal and vertical wind components, respectively, $\nabla_h$ the horizontal gradient, $\kappa = \frac{R}{c_p} = 0.286$, and $p$ pressure with $p_0$ the reference pressure of 1000 hPa. The first three terms on the rhs represent the contributions (i) to (iii), while the fourth term is a residual arising mainly from temporal changes of the climatology, e.g., the daily cycle and seasonality, and the fact that $T'$ is never exactly 0 K at the time of anomaly genesis. In general, this term is small compared to the other terms (Fig. 9 and Röthlisberger and Papritz, 2023b). Based on contributions of the processes (i)–(iii), the trajectories were further categorized as (a) advective only (the advective term is negative, but adiabatic and diabatic terms are positive), (b) advective dominant (the advective term is at least twice as large as the diabatic and adiabatic terms), (c) advective and diabatic (both are negative, but neither of them is twice as large as the other), (d) diabatic dominant (the diabatic term is at least twice as large as the advective and adiabatic terms), (e) other (all other possibilities). Supplementary Fig. 1 illustrates how the trajectory analyses were integrated with the 850 hPa temperature and circulation SOM analyses.

### 2.2.4 Long-term changes

Long-term changes in cold anomaly occurrence were analyzed using linear trends during the period 1979–2020. The trend calculations were based on the annual occurrences of each SOM node. The F statistics were applied to test whether the trends were statistically different from zero, having 0.05 as a threshold for the p-value. Long-term changes in the origin and physical





processes contributing to cold anomalies were assessed by comparing results for the first (1979–1999) and second half (2000–

2020) of the study period.

## 3 Results

### 3.1 Temperature anomalies over Europe

In general, the strength of the cold 850 hPa temperature anomalies over Europe in winter is largely linked to distance from the

sea; however, the Black Sea and the Baltic Sea seem to be too small and cold to reduce the strength by significantly warming the lower troposphere up to the 850-hPa level. The coldest European anomalies are found over southeastern Europe, with the 10[th] percentile below -7°C (Fig. 1). The weakest cold anomalies, with the 10[th] percentile being approximately -5°C (Fig. 1), are found over the Mediterranean and along the North Atlantic coast. During the latter period (1979–1999), the coldest days (10[th] percentile) became 0.4–1.6 K warmer over the eastern and northern parts of Europe compared to the first period (2000–

2020) but remained the same in Western Europe (not shown). Interestingly, over Eastern Europe, the lowest winter 850 hPa temperatures (10[th] percentile) warmed less than the median and highest (90[th] percentile) winter temperatures (not shown).

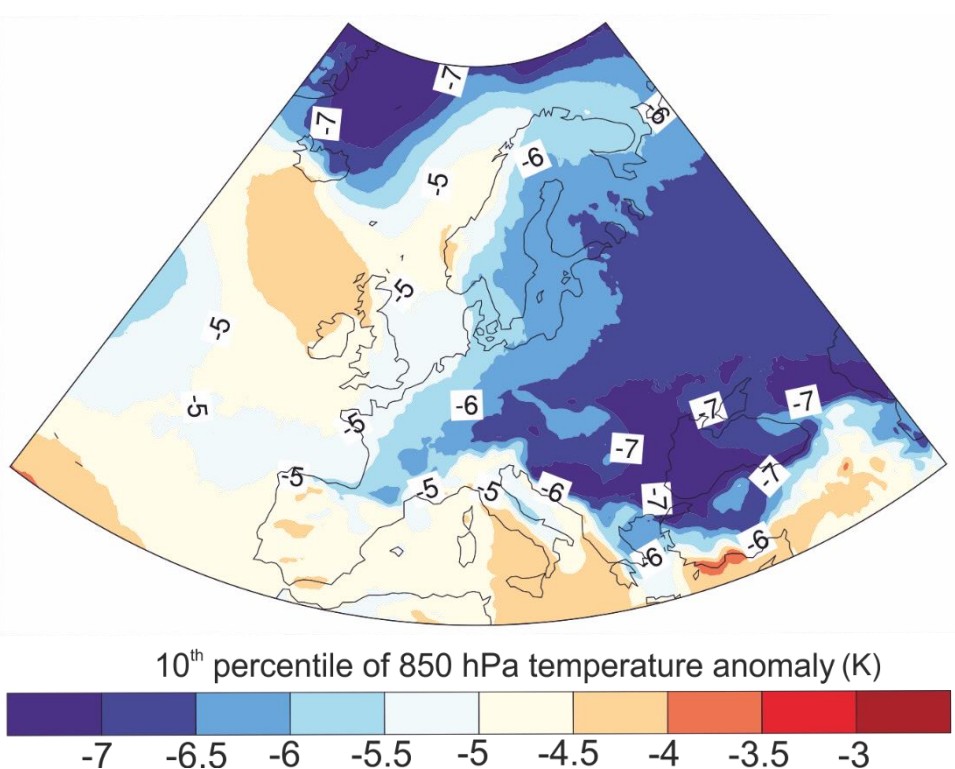

**Fig. 1 10[th] percentile of 850 hPa temperature anomalies (i.e., coldest anomalies) in November–March 1979–2020.**

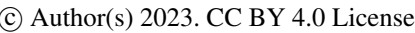


The variety of daily 850 hPa temperature fields is represented by 56 characteristic temperature anomaly fields over Europe allocated using the SOM approach (Fig. 2). A prominent feature in all the fields (Fig. 2) is that there are simultaneously both warm and cold anomalies over Europe and the northern North Atlantic. This means that a cold anomaly with respect to the climatology is practically always present somewhere over Europe, having also its counterpart, a warm anomaly; this is typically

related to waviness of the Polar front jet stream. The 850 hPa temperature anomalies over Europe have also clear connections with anomalies outside Europe, particularly over Greenland, the Barents Sea and Ural region (Supplementary Fig. 2).

We limit the rest of the analysis to the 14 anomaly fields in Fig. 2, in which the composite cold anomaly below the 10th percentile is shown in Fig. 1 (see description in Sect. 2.2b). Of those, six main cold anomaly types were formed: cold southeast

(Type I), cold extended southeast (Type II), cold central and west (Type III), cold central, west and north (Type IV), cold north (Type V) and cold Europe (Type VI), shown in the first column of Fig. 3. The types were formed based on the location of the temperature anomaly colder than the 10th percentile. The number of cases belonging to these types is listed in Table 1. Generally, there has been a linear trend of -2.7 days decade$^{-1}$ in the number of days having any of the cold anomaly Types I–VI. In the following, we go through characteristics of these cold anomaly types one by one. For more detailed results,

Supplementary Fig. 7 can be used as a reference to make connections between the cold anomaly types (the left column in Fig. 3), individual circulation types (Supplementary Figs. 3 and 4) and climate indices (Supplementary Fig. 5).

### 3.2 Cold anomaly types

### 3.2.1 Cold southeast (Type I)

Type I is formed of nodes 22 and 29, which both have temperatures below the 10th percentile over southeastern Europe.

Including areas of less extreme anomalies, the entire cold anomaly reaches from the Balkan Peninsula to Caucasus, and a strong, warm counterpart is located over northern Europe (Figs. 2 and 3). This cold anomaly occurs simultaneously with a cold anomaly over Greenland (Supplementary Fig. 2) and is most common in mid-winter, during December–February.






**Fig. 2 A 7 x 8 SOM composite array of daily 850 hPa temperature anomalies in November–March 1979–2020. The six cold anomaly types (I-VI) are marked.**



**Table 1** Details of the six cold anomaly types. The node numbers refer to Fig. 2, N is the number of cases belonging to a node/anomaly type, N (trajectory) indicates the number of cases for which the trajectory analyses were made, the number of trajectories indicates the size of the trajectory ensemble linked to all cases of a node/type, and linear trends indicate the trend during 1979–2020 for all N cases. The statistically significant trends at the 95% confidence level are marked bold.

| | Node | N | N (trajectory) | Number of trajectories | Linear trend (days decade$^{-1}$) |
|---|---|---|---|---|---|
| **Cold southeast** | 22 | 112 | 109 | 114077 | 0.08 |
| **(Type I)** | 29 | 121 | 118 | 94615 | -0.11 |
| | ALL | 233 | 227 | 208692 | -0.03 |
| **Cold extended southeast** | 36 | 130 | 121 | 119365 | -0.68 |
| **(Type II)** | 43 | 96 | 91 | 80946 | -0.24 |
| | 50 | 180 | 177 | 228670 | -0.28 |
| | ALL | 406 | 389 | 428981 | -1.20 |
| **Cold central and west** | 45 | 85 | 81 | 45170 | 0.01 |
| **(Type III)** | 52 | 139 | 121 | 78545 | 0.54 |
| | ALL | 224 | 202 | 123715 | 0.55 |
| **Cold central, west and north** | 53 | 107 | 103 | 115697 | 0.00 |
| **(Type IV)** | 54 | 108 | 107 | 157910 | 0.33 |
| | ALL | 215 | 210 | 273607 | 0.33 |
| **Cold north** | 47 | 97 | 90 | 91413 | -0.63 |
| **(Type V)** | 48 | 83 | 76 | 97234 | -0.42 |
| | 49 | 129 | 125 | 175126 | **-0.86** |
| | ALL | 309 | 291 | 363773 | **-1.91** |
| **Cold Europe** | 55 | 152 | 150 | 172798 | 0.04 |
| **(Type VI)** | 56 | 113 | 111 | 175873 | -0.44 |
| | ALL | 265 | 261 | 348671 | -0.40 |
| **All cold anomaly types** | ALL | 1652 | 1580 | 1747439 | -2.67 |


The cold anomaly Type I is typically preceded by a block over central Europe (Supplementary Figs. 4 and 7), which enables transport of cold air from the north towards southeastern Europe, along the western edge of a trough located over Russia (Fig. 4). As a part of the dynamical development, the MSLP high moves farther south, strengthening the westerly flow to eastern Europe but sheltering southeastern Europe from the warm westerly flow (Fig. 3). Comparison of the MSLP and 500 hPa

geopotential fields indicates that the structure is more baroclinic in the eastern parts of the study region, reflected in the strong temperature gradient between Scandinavia and the Black Sea region (Figs. 3 and 4). Generally, these circulation features are linked to mostly positive phases of NAO and AO, and to low values of GBI (Supplementary Figs. 5 and 6). An interesting methodological aspect is that the composite of all MSLP fields linked to this cold anomaly type (second column in Fig. 3) suggests that the high pressure is located over eastern Europe, whereas the most common individual MSLP patterns display





highest MSLP in the southwestern Europe (third and fourth columns in Fig. 3). This clear difference indicates that the MSLP composite does not well represent the individual circulation types.

Initially, 10 d before, air masses linked to Type I cold anomalies were roughly 1 K colder than the climatological mean of the corresponding trajectory point (Fig. 5); 6 K cooling compared to the surrounding climate started 5 d before the arrival. The

trajectory ensemble indicates that 5 d before arrival to southeastern Europe the air parcels were most commonly found over the Norwegian Sea, Scandinavia, and Eastern Europe (Fig. 6a). More than 60 % of the anomalies belonging to Type I were formed due to advection only or advection being at least twice as large as diabatic and adiabatic contributions (Fig. 7). 26% of the cases were defined as 'advective and diabatic', meaning that there was notable diabatic cooling, the contribution of which was comparable to the contribution of cold advection. In a Ɵ–T diagram, the median of all cold anomaly Type I trajectories

shows a combination of diabatic cooling and adiabatic warming during the early days, and mostly adiabatic warming during the last days before the arrival (Fig. 8). However, when trajectories dominated by different processes are viewed separately (third row in Fig. 8), it is clear that very different combinations of processes can lead to Type I cold anomalies with an equal magnitude. Diabatic-dominant trajectories experienced a large drop of potential temperature along the path, whereas the potential temperature of advective-only trajectories increased along the path, indicating diabatic warming (Fig. 8). The cold

anomaly genesis points for the advective-dominant trajectories were located over Eastern and Northern Europe and over the Barents and Greenland Seas (Supplementary Fig. 8). The diabatic-dominant trajectories originated from farther south, from south-eastern Europe. Generally, nearly all the trajectories experienced strong adiabatic warming, but its effect was overcompensated by horizontal advection and partly also by diabatic cooling (Fig. 9).

The annual occurrence of Type I did not have any clear linear trend during 1979–2020 (Table 1). However, during the latter period (2000–2020) air parcels more commonly originated from continental Europe than from the Norwegian Sea (Fig. 6 b). In addition, the initial temperatures were almost 1 K higher during the latter period, but at the end point (at 0 h) the temperature with respect to climatology was only 0.5 K higher during the latter period (Fig. 5). This seems to be due to reduced adiabatic warming during the latter period (Fig. 9). During 2000–2020, cold anomaly Type I was associated with very positive NAO

and AO indices, very low GBI, and a negative anomaly in Ural MSLP (Supplementary Figs. 5 and 7), while during 1979–1999 the signals were not as strong.

### 3.2.2 Cold extended southeast (Type II)

Type II, consisting of nodes 36, 43 and 50, has temperatures below the 10[th] percentile over the Balkans. The whole cold anomaly covers a large area of the southeastern side of Europe, reaching from Italy to western Russia (Figs. 2 and 3), and there

is a simultaneous cold anomaly over Greenland like in Type I (Supplementary Fig. 2). The warm counterpart is found over the Norwegian Sea, the Nordic countries and British Isles. This cold anomaly type occurs quite evenly during all months of November–March.



The cold anomaly Type II is typically preceded by a block over western Europe (Supplementary Figs. 4 and 7), like cold anomaly Type I, but in Type II the MSLP high moves over Scandinavia and western Russia, thus to a more northeastern location than in Type I (Figs. 3–4). The associated positively (i.e., from northeast to southwest) tilted trough at 500 hPa is located over the similarly tilted cold anomaly (Fig. 4). The MSLP composite for Type II (the second column in Fig. 3) is very similar to that for Type I, and suggests that a high pressure centered over central Europe would be the dominant feature with the cold anomaly forming on its eastern flank. However, the most common individual MSLP patterns are distinct from Type I: the most frequent MSLP patterns during this cold anomaly type have a strong high pressure over western Russia and Scandinavia (Fig. 3), sheltering eastern and southeastern Europe from the warm westerly flow from the northern North Atlantic, making the cold anomaly region to reach farther northwest than in Type I. The cold anomaly Type II mostly occurs during the positive phase of NAO and in the presence of UH, whereas AO can be either positive or negative (Supplementary Figs. 5 and 7). This anomaly type is not sensitive to the strength of GBI.

Air parcels linked to cold anomaly Type II mainly originated from the same regions as in Type I, namely from the Norwegian Sea and Scandinavia, but the origins were partly spread out farther to the east (Fig. 6). Initially, 10 d before, the air parcels were 1 K colder than the surrounding climatology. The period over which the anomaly formed lasted two days longer than for Type I, and started already 7 d before the arrival. The final temperature anomaly was -8 K. Roughly 50% of the cold anomalies belonging to Type II were due to advection only or advection dominant, whereas approximately 30% were due to both cold advection and diabatic processes. The median of all trajectories seen in a Ө–T diagram indicates a smooth descent from 700 hPa with adiabatic warming and simultaneous diabatic cooling (Fig. 8). The evolution of trajectories plotted separately for different processes resembles the evolution of Type I trajectories. Cold anomalies forming due to advection typically had their genesis point over northern Europe or Norwegian or Barents Seas, thus in a colder climate (Supplementary Fig. 8). The diabatic dominant trajectories, in turn, mainly originated from the eastern half of continental Europe, where the cold land surface allowed for efficient radiative cooling, and the diabatic heating at the surface was lacking. As in Type I, cold advection and diabatic cooling generally overcompensated adiabatic warming (Fig. 9). However, 30% of the trajectories experienced adiabatic cooling due to ascent of the air parcels to 850 hPa (Fig. 9), and the adiabatic process thus strengthened the cold anomaly.






**Fig. 3 Composites of 850 hPa temperature in the six cold anomaly types (first column on the left), composites of MSLP in each of those cold anomaly types (second column), the most common MSLP circulation type during each cold anomaly type (third column), and the second most common circulation type during each cold anomaly type (fourth column). Each row shows results for a certain**

**cold anomaly type. The numbers in the third and fourth columns indicate the MSLP SOM node number (see Supplementary Figure 3) and the percentage of cases associated with this circulation type in each cold anomaly type.**



**Fig. 4 Composites of 850 hPa temperature in the six cold anomaly types (first column on the left), composites of 500 hPa geopotential**
**in each of those cold anomaly types (second column), the most common 500 hPa geopotential pattern during each cold anomaly type**
**(third column) and the second most common 500 hPa geopotential pattern during each cold anomaly type (fourth column). Each**
**row shows results for a certain cold anomaly type. The numbers in the third and fourth columns indicate the circulation SOM node**
**number (see Supplementary Figure 3) and the percentage of cases associated with this circulation type in each cold anomaly type.**






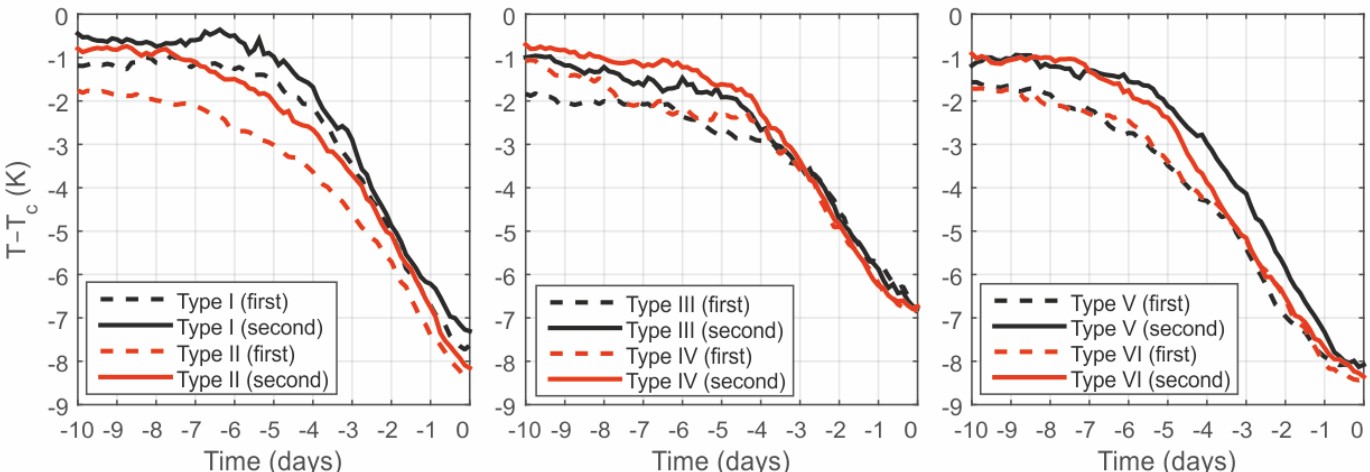

**Fig. 5 Median temperature anomaly with respect to the climatological mean temperature of the trajectory point Tc, based on the full trajectory ensemble. Time of the backward trajectories is on the x-axis. The dashed lines show results for 1979–1999 (first period) and solid lines for 2000–2020 (second period).**

The annual occurrence of cold anomaly Type II had a decreasing linear trend (-1.2 days decade$^{-1}$) during the study period (Table 1), but the trend was not statistically significant. During 1979–1999, the Ural MSLP anomaly associated with Type II

events was typically positive, even strongly positive, but during 2000–2020 also negative UH anomalies occurred (Supplementary Figs. 5 and 7). Furthermore, during the latter period, the air masses more commonly originated from the Norwegian Sea and less commonly from continental Europe (Fig. 6b). The initial cold temperature anomaly with respect to the climatology decreased by 1 K compared to the first period (Fig. 5), but this was compensated by an increase in diabatic cooling along the trajectory during the second period (Fig. 8). Consequently, at the end point (at 0 h) the temperature anomalies

were almost equal during the two periods (Fig. 5).

### 3.2.3 Cold central and west (Type III)

Type III, consisting of nodes 45 and 52, has an extensive cold anomaly in the central and western parts of Europe (Figs. 2 and 3), having temperatures below the 10$^{th}$ percentile over a small area over Switzerland, Austria and northern Italy. The cold anomaly is located between two warm anomalies, one of which is located over the Norwegian Sea and Greenland and the other

one over western Russia (Supplementary Fig. 2). The cold anomaly Type III is most common in December.

The cold anomaly Type III is typically preceded by a cold-air outbreak from the northwest (Fig. 6, Supplementary Figs. 4 and 7). During the cold anomaly, there is a ridge at 500 hPa blocking the westerly flow; the ridge is located west of the British





Isles (Fig. 4), thus farther west than during cold anomaly Types I–II. Otherwise, the 500 hPa geopotential gradients are very weak, indicating a weak zonal flow (Fig. 4). MSLP is generally high over Europe (Fig. 3). The dominant circulation types

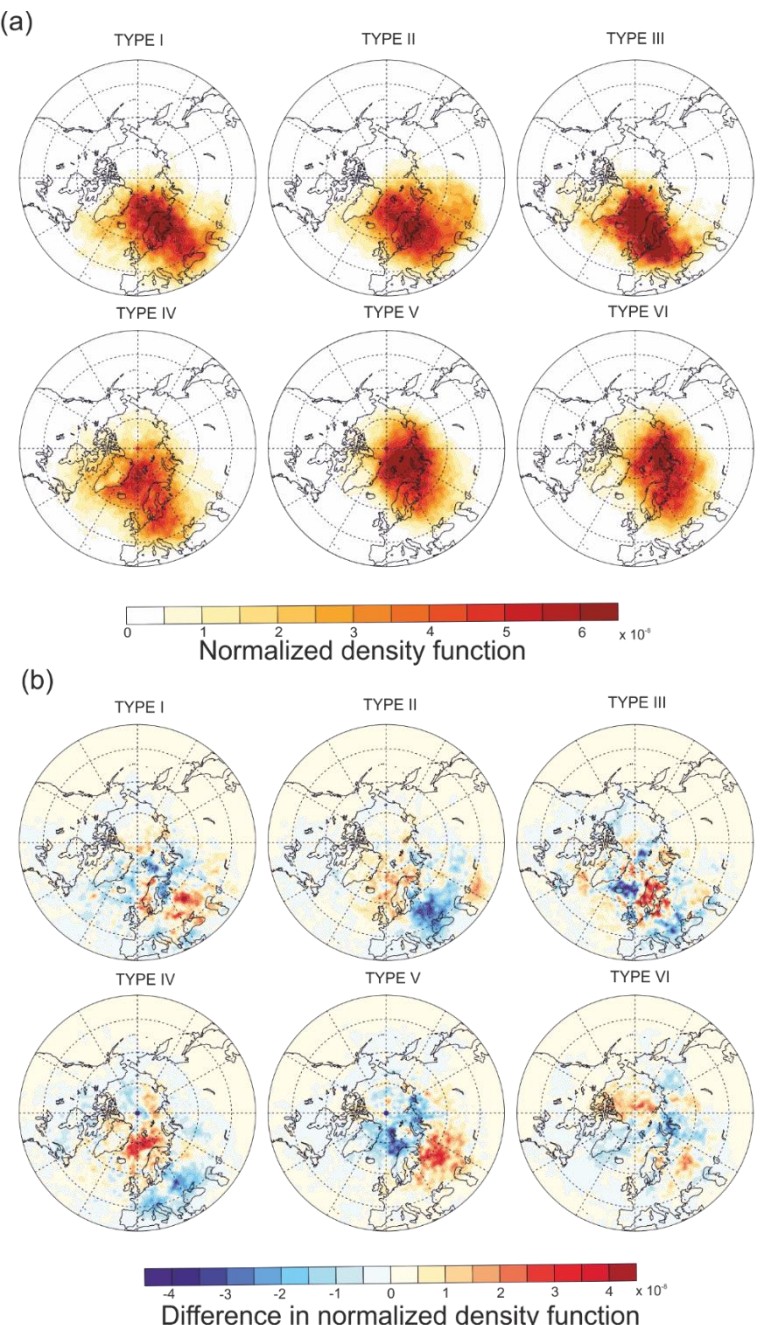

**Fig. 6 (a) Normalized density of trajectory ensemble start points, 5 days before, for the six cold anomaly types. (b) Difference in normalized density of trajectory ensemble start points, 5 days before, between the second period (2000–2020) and the first period (1979–1999).**



have a strong high in MSLP reaching from western Russia to Scandinavia or even over the British Isles. The composite of MSLP (the second column in Fig. 3) has features of the high pressure areas over Russia and the British Isles, but depicts them as separate high pressures. Both the weak 500 hPa geopotential gradients and high MSLP enable the cold air mass to stay over central Europe without being rapidly advected elsewhere. The cold anomaly Type III is not sensitive to the phase of NAO; both positive and negative indices occur prior to and during the events (Supplementary Figs. 5 and 7). AO is most commonly negative, GBI typically high, and the MSLP anomaly over the Urals more often positive than negative.


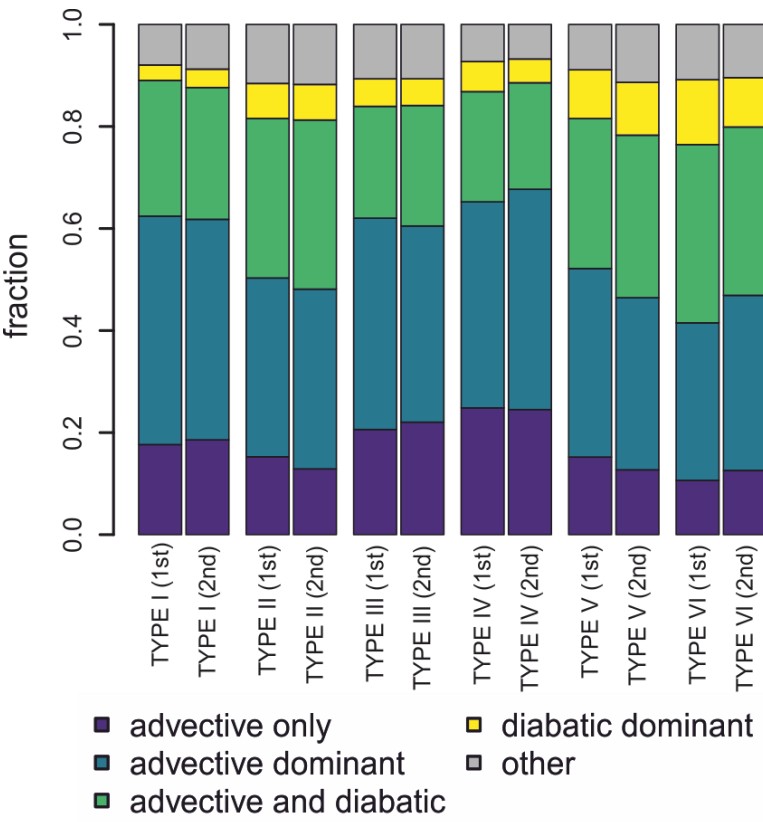

**Fig. 7 Fraction of trajectories belonging to categories of advective only, advective dominant, advective and diabatic, diabatic dominant and other (see Sect. 2.2.3) in each of the six cold anomaly types. The left bar shows results for 1979–1999 (1st period) and the right bar for 2000–2020 (2nd period).**




**Fig. 8 (a) Ѳ–T diagrams showing the temporal evolution of the medians of Ѳ and T along 10-d trajectories associated with the six cold anomaly types for (a) the two periods (1979–1999 and 2000–2020) and (b) the subcategories based on the physical processes causing the final temperature anomaly at 850 hPa (see Sect. 2.2.3). Note that isobaric temperature changes purely caused by diabatic heating/cooling are associated with movement along the diagonal lines, whereas purely adiabatic temperature changes are associated with movement parallel to the x-axis.**



Due to the northerly flow preceding the cold anomaly Type III, the air parcels mostly originated either from the Northern Europe or the Norwegian Sea, commonly also from the east coast of Greenland. In consequence, along most of these trajectories (60%) the cold anomalies were only or dominantly caused by advection (Supplementary Fig. 8). Furthermore, 42% of the trajectories experienced diabatic heating, which was linked to the large upward sensible heat flux due to the relatively
warm sea surface compared to the air temperature (Fig. 9). Only for a small fraction (~5%) of trajectories, diabatic cooling was the dominant process causing the temperature anomaly, with their genesis point located over continental Europe (Supplementary Fig. 8). The air parcels cooled with respect to the surrounding climate along the 10-d trajectory; the cooling was most efficient during the last 3–4 d before the arrival (Fig. 5). The median trajectory in the Ө–T diagram indicates a slight diabatic cooling, which is overcompensated by smooth, but strong adiabatic warming along the trajectory (Fig. 8). When
viewed separately, the air parcels linked to advection-only trajectories increase their temperature and potential temperature along the pathway, suggesting that they move at low altitudes over the open ocean. Advective dominant trajectories, in turn, typically increase their temperature due to adiabatic warming, but simultaneously decrease their potential temperature due to radiative cooling (Fig. 8). The trajectories with a clear diabatic contribution show a large drop in potential temperature, but a minor increase in temperature along the pathway. Hence, both initially colder and warmer air parcels compared to the final
temperature (0 h) could lead to cold anomalies in the central and western parts of Europe.

There has been a little, but statistically insignificant, increase (0.55 days decade$^{-1}$) in the annual occurrence of this cold anomaly type during 1979–2020 (Table 1). No clear or systematic circulation changes linked to Type III were detected (Supplementary Fig. 7), but during the latter period, the air masses less commonly originated from the Norwegian Sea, and more commonly from Scandinavia (Fig. 6b). The initial cold temperature anomaly with respect to the climatology was 1 K smaller during the
latter period, but the end point (0 h) temperature anomaly remained the same (Fig. 5).

### 3.2.4 Cold central, west and north (Type IV)

Type IV, consisting of nodes 53 and 54, has an extensive cold anomaly over western, central and northern Europe (Figs. 2 and 3); the mean 850 hPa temperature is below the 10th percentile over France. The cold air mass is located between two warm counterparts, one located over Greenland and the other one near the Black Sea (Supplementary Fig. 2). Type IV is frequent
throughout winter, being most common in January and February.

Type IV is most commonly linked to circulation patterns having a low in MSLP and a clear 500 hPa trough over Scandinavia (Figs. 3 and 4). In addition, there is typically a 500 hPa ridge over the northern North Atlantic, which has thus a more western location than in cold anomaly Types I–III. All of these features indicate particularly meridional circulation from the north. Nearly as frequently, the cold anomaly Type IV is associated with a blocking over the British Isles (circulation nodes 15 and
20 in Supplementary Figs. 3, 4 and 7), which prevents the westerly flow to reach the cold anomaly region, but is not associated with a strong northerly flow. For this cold anomaly type, the circulation composites agree fairly well with the most common





circulation patterns (Figs. 3 and 4). Cold anomaly Type IV is associated with negative NAO and AO indices together with a strong Greenland blocking both during and prior to the events (Supplementary Figs. 5 and 7).


**Fig. 9 Percentage contributions of adiabatic process, diabatic processes, horizontal advection and the residual term to the final 850 hPa temperature anomaly in each of the six cold anomaly types. In each plot, the left bar shows results for 1979–1999 (first period) and the right bar for 2000–2020 (second period).**

The air parcels of cold anomaly Type IV typically originated from the northern North Atlantic and Northern Europe, similarly as in Type III (Fig. 6). The strong meridional flow linked to the MSLP low and 500 hPa through favored trajectories with the origin in the northern North Atlantic, and for these trajectories the advection was the dominant process (Supplementary Fig. 8). Two thirds of the trajectories had advection as the only or dominant process to cause the cold anomaly, and the fraction was highest of all the cold anomaly types (Fig. 7). The block over the British Isles favored origins closer to the cold anomaly

region; the trajectories, which were diabatic dominated, originated from a close distance around the cold anomaly region (Supplementary Fig. 8). Median initial temperature deviation from the climatology was 1 K, and the largest temperature drop





compared to the surrounding climatology occurred during the last 4 days before the arrival (Fig. 5). The Ө–T diagram indicates a two phase evolution, namely a first one during the early days of the trajectory when most of the diabatic cooling occurs, and a second after that where median Ө stays nearly constant but T increases due to adiabatic warming (Fig. 8). The early, strong

diabatic cooling is seen in all the trajectory types with advection playing a major role, but for the diabatic-dominant trajectories efficient radiative cooling occurs throughout the trajectory evolution (Fig. 8). However, not all the trajectories cooled diabatically; 43 % of the Type IV trajectories experienced diabatic warming, which was the highest proportion of all the cold anomaly types (Fig. 9). Furthermore, 75% of the trajectories experienced adiabatic warming (Fig. 9). The warming was overcompensated by the strong cold advection, which was a particularly important process for the formation of Type IV cold

anomalies.

The annual occurrence of Type IV did not have any clear linear trend during 1979–2020 (Table 1), and no obvious circulation changes were detected (Supplementary Fig. 7). In addition, the initial temperature anomaly and the temperature anomaly at the end point (at 0h) remained nearly the same with respect to the climatology (Fig. 5). However, during the latter period, the trajectories more commonly originated from the northern North Atlantic, and less commonly over the continental Europe (Fig.

6b). This was also seen in a larger fraction of advective dominant and a smaller fraction of diabatic dominant trajectories during the latter period (Fig. 7). Adiabatic warming was stronger during the latter period, and it was compensated by stronger cold advection (Fig. 9).

### 3.2.5 Cold north (Type V)

Type V, consisting of nodes 47, 48 and 49, has 850 hPa temperatures below the 10th percentile over Scandinavia and the Baltic

countries, and a weaker cold anomaly extends from Scandinavia either to western or eastern Mediterranean, depending on the node (Figs. 2 and 3). Warm counterparts are found over Greenland/Canadian Arctic Archipelago and over the region from Turkey to the Caspian Sea, or even farther east (Supplementary Fig. 2). This anomaly type is most common in November and January.

Before and during the cold anomaly Type V, there is typically a 500 hPa ridge located either over the British Isles or

Scandinavia, blocking the air masses from the northern North Atlantic. However, the westerly flow is not blocked in all of the preceding circulation patterns (Supplementary Figs. 4 and 7). Type V is linked to mostly similar dynamical features as Type IV, but more often associated with lows in MSLP and 500 hPa centered south of Iceland, instead of Scandinavia as in Type IV (Figs. 3 and 4). The composite MSLP displays very weak pressure gradients over Europe (Fig. 3), but this seems to be a result of averaging distinct, compensating MSLP patterns rather than representing the prevailing dynamical conditions in

individual cases. In fact, the most common individual pressure patterns have much stronger pressure gradients, but a lot of variability between the patterns. Events of Type V mostly occur under negative NAO and AO. In addition, there is a strong Greenland blocking and UH, between which the cold air mass is located (Supplementary Figs. 5 and 7). The negative NAO does not itself bring cold air but allows UH to steer cold air to northern Europe.



The air parcels linked to cold anomaly Type V commonly originated from the Barents and Kara Seas, Scandinavia and northern
coastal land of Russia west of 90°E; thus, clearly from a more eastern region than in cold anomaly Types I–IV (Fig. 6a). Driven
by the circulation patterns, many of the trajectories made a detour to the south over Russia before reaching Scandinavia from
the southeast (not shown). There was a clear distinction that the trajectories linked to cold anomalies due to advection
originated from the Arctic sea areas, whereas cold anomalies due to diabatic processes had their genesis point over the above-
mentioned land areas (Supplementary Fig. 8). Compared to Types I–IV, the proportion of cold anomalies formed due to
diabatic processes was larger, being still only 8 % (Fig. 7). Furthermore, diabatic processes contributed by a comparable
amount with cold advection in 31% of the trajectories (Fig. 7), and also this proportion was clearly higher than in Types I, III
and IV. The trajectories mainly cooling due to diabatic processes experienced a steady cooling along the pathway. However,
the trajectories, in which mainly advection accounted for the cold anomaly, experienced strong diabatic cooling during the
early days of the trajectory, after which they adiabatically, and partly also diabatically, warmed during a descent to 850 hPa
(Fig. 8); this behavior was different from Types I–III which did not have a clear initial diabatic cooling period. In general, the
relative role of diabatic cooling was larger in Type V than in Types I–IV.

There has been a statistically significant decrease (-1.9 days days decade$^{-1}$) in the annual occurrence of this cold anomaly type
during 1979–2020 (Table 1). In particular, during 2000–2020, the trajectories less frequently originated from the Barents and
Kara Seas compared to the earlier period (Fig. 6b); these sea areas have been exposed to the most extreme climate warming
on the Earth (Rantanen et al., 2022) and, therefore, might have become less significant sources of cold air. However, the
trajectories more commonly originated from western Russian land areas during the latter period, which was seen as an increase
of the fraction of diabatic dominated trajectories. Interestingly, the shift of trajectory origins from ocean to land was the
opposite to the shift seen in Type IV. During the latter period, Type V cold anomalies were more commonly preceded by
blocking around the anomaly region; during the earlier period, even a combination of positive NAO and AO without Greenland
blocking and UH could precede the cold anomaly (Supplementary Figs. 5 and 7).

### 3.2.6 Cold Europe (Type VI)

Type VI consists of nodes 55 and 56, in which 850 hPa temperature is below the 10$^{th}$ percentile over the British Isles, southern
Scandinavia and Baltic countries; the whole cold anomaly extends over most of Europe, except the southeastern corner (Figs.
2 and 3). A warm counterpart is found over Greenland (Supplementary Fig. 2). This cold anomaly type is most common in
January–March.

As a prominent feature, the most common circulation types linked to Type VI are associated with very weak 500 hPa
geopotential gradients over the northern North Atlantic (Fig. 4). In particular, 500 hPa geopotential pattern in node 15 over the
northern North Atlantic resembles a Rex block (Sousa et al., 2021) isolated from the subtropical belt (Fig. 4). These features
indicate a very weak jet stream, or even partly westward flow at 500 hPa, which limits access of warm and moist air masses
from the northern North Atlantic. However, the westerly flow is not blocked in all of the preceding circulation patterns



(Supplementary Figs. 4 and 7). Another common denominator for the 500 hPa patterns is a trough over Scandinavia and the Barents Sea. The MSLP fields show a high extending from Greenland's coast to Scandinavia and farther southeast (Fig. 3). In the composite MSLP, the high pressure is visible over the Norwegian Sea and Scandinavia, but the pressure gradients are generally weak, whereas the most common individual MSLP patterns have much stronger pressure gradients. Type VI is
related to mostly negative phases of NAO and AO, but occasionally preceded by a period of positive phases of NAO and AO (Supplementary Figs. 5 and 7). In addition, the cold air mass is located between a Greenland block and Ural high (Supplementary Figs. 5 and 7).

Air parcels of Type VI most commonly originated from the Kara Sea, but often also from the Barents Sea, Scandinavia and
northern coastal land of Russia west of 90°E. The cold anomalies linked to air parcels originating from the Arctic seas were mainly due to advection, whereas diabatic processes dominated for air parcels originating from the land areas (Supplementary Fig. 8). The cooling period with respect to the climatology was equally long as in Type II, and thus longer than in the other types, starting already 7 d before the arrival (Fig. 5). This was probably linked to the generally weak flow, which made the travel time from the anomaly source region relatively long, also giving time for diabatic cooling. The proportion of cold
anomalies formed mostly due to diabatic processes was the largest, 9%, of all the six types (Fig. 7). Furthermore, diabatic processes contributed by a comparable amount with cold advection in 34% of the trajectories (Fig. 7). Hence, diabatic cooling played the largest role for Type VI of all the types (Figs. 7 and 9). Median trajectories in the $\Theta$–$T$ diagram resemble those of Type V; the trajectories, in which mainly advection was the reason for the cold anomaly, experienced strong diabatic cooling during the early days of the trajectory, after which they adiabatically, and partly also diabatically, warmed during a descent to
850 hPa (Fig. 8).

The annual occurrence of Type VI did not have any clear linear trend during 1979–2020 (Table 1), and no clear circulation changes were detected (Supplementary Fig. 7). The role of diabatic cooling decreased during the latter period, whereas the contribution of cold advection increased (Figs. 7 and 9); this was probably due to trajectories more commonly originating from the sea ice covered Arctic Ocean during 2000–2020.

**4 Discussion**

In many previous studies, cold extremes over Europe have been related to the NAO index. Can a simple index, like NAO, describe dynamical and physical mechanisms linked to cold anomalies in sufficient detail? Our results indicate that cold anomalies over northern, western and the entire Europe (Types IV–VI) are linked to the negative phase of NAO, whereas cold anomalies over southeastern Europe (Types I and II) are connected to the positive phase of NAO. Previously, low wintertime
(2-m) temperatures in southern (in particular southeastern) Europe have been in some studies attributed to the positive phase of NAO (Hurrell, 1995; Castro-Díez et al., 2002; Trigo et al., 2002; Giorgi and Lionello, 2008), but associations have also been found between negative NAO and cold winter weather in southern Europe (Castro-Díez et al., 2002; Anagnostopoulou



et al., 2017). Overall, the effect of the NAO on winter temperatures in southern Europe has been found to be nonsystematic, varying from year to year and being sensitive both to the exact location of the NAO centers of action and to the specific location
for which the effects are studied (Castro-Díez et al., 2002; Giorgi and Lionello, 2008). There are some indications of such a high sensitivity even in northern Europe (Koslowski and Loewe, 1994). The impact of NAO may depend on its exact definition. Early studies defined NAO on the basis of the meridional pressure gradient between Iceland and Azores (Barnston and Livezey, 1987), whereas the Principal Component -based NAO is calculated on the basis of the time series of the leading Empirical Orthogonal Function of MSLP anomalies over the Atlantic sector, 20°-80°N, 90°W-40°E (Hurrell et al., 2003). The latter
approach accounts also for the almost symmetric (in North-South and West-East directions) patterns we identified applying SOM. All the above suggests that the complex teleconnections are not adequately represented by the NAO, especially by a simple NAO index based on a meridional pressure gradient. The SOM analyses, in contrast, provide a remarkable added value by revealing the large diversity of large-scale circulation patterns that can cause cold extremes over Europe despite a similar NAO index.

Previous studies have indicated that a high GBI is associated with cold anomalies in Northern, Western and Central Europe (Vihma et al., 2020; Hanna et al., 2022), which is confirmed in our study (seen in Types III–VI). Based on our results, we see that a high GBI is an indicator for blocking of the westerly flow from the northern North Atlantic, enabling cold advection from climatologically colder regions and efficient radiative cooling over eastern land areas along the pathway. Cold anomalies in eastern Europe have previously been found to be associated with strong UH, especially when it is associated with an upper-
level Rossby wave breaking over Scandinavia (Sui et al., 2022). Sui et al. (2022) emphasized the role of strong advection of cold air masses for causing cold anomalies linked to strong UH. However, our cold anomaly Types V and VI, associated with a strong UH, indicate that even if the role of cold advection is large for these anomalies, it is noteworthy that the relative role of diabatic cooling was larger for these types than for the other cold anomaly types in Europe. This was due air parcels originating from the Barents and Kara Seas, Scandinavia and northern coastal land areas of Russia, allowing for more efficient
radiative cooling over the cold land surface and the absence of diabatic heating along their pathway compared to air masses originating from farther west in Types I–IV, which partly move over open ocean.

Our results confirm the fundamental role of blocking in transporting cold air from climatologically colder regions, as earlier shown by Trigo et al. (2004), Sillmann et al. (2011), Pfahl (2014) and Sousa et al. (2018). Our results give evidence that it is beneficial to detect blocks without any predefined region, as the cold anomaly region moves eastward with the block located
farther east. This is seen when looking at the locations of the block and the cold anomaly, starting from Type IV to Type I (Fig. 4). However, Type V and VI anomalies are found in between a strong Greenland blocking and Ural high. Related to this division, there was also a clear distinction that the air parcels of Types I–IV mainly originated from the northern North Atlantic, Scandinavia and eastern Europe, whereas the origins of Types V and VI (the latter being a European wide cold anomaly) were over the Barents and Kara Seas, Scandinavia and northern coastal land of Russia.





We also found that the six cold anomaly types differed from each other in terms of the relative importance of cold air advection, diabatic heating/cooling, and adiabatic warming/cooling. The role of cold-air advection (as the single or dominating process contributing to the cooling) varied between the types, being largest for Types I, III and IV (Fig. 7), whereas cold anomaly types most affected by radiative cooling were V and VI. The cold anomalies mainly due to advection originated from the Arctic sea regions (except in Type I). The cold anomalies dominated by diabatic processes originated from more southern, continental

origins, closer to the actual European regions of cold anomalies. These results, indicating that anomalies over western Europe and southeastern Europe are predominantly caused by advection, and over eastern Europe by both advective and diabatic processes, are well in line with Röthlisberger and Papritz (2023) who addressed physical processes causing 2-m temperature cold anomalies. Our results also agree with Bieli et al. (2015) in that the air temperature along the trajectories is often lowest in the source region but the extremeness of the air mass with respect to its local surroundings increases along the pathway.

However, when the trajectories were divided into subcategories based on the dominating physical processes, we found that potential temperature often drops due to effective radiative cooling along the trajectory, while the temperature usually increases as subsidence warming dominates over diabatic cooling. A drop in temperature is sometimes also seen (Fig. 8).

Our results are in line with Smith and Sheridan (2020), who found that cold air outbreaks in Europe have remained as extreme and nearly as common as decades ago. We only detected a minor decrease (1% decade$^{-1}$, Table 1) in the occurrence of cold

extremes in Europe, although the climate has notably warmed during the last 40 years, particularly in the Arctic. Why have the cold-air outbreaks hitting Europe not changed? It seems to be partly due to enhanced diabatic (radiative) air-mass cooling along the pathway from the site of air mass origin to Europe, which has compensated for the warming in the site of origin. The more efficient radiative cooling is most probably related to the generally higher radiative cooling rate of warmer and moister air. Furthermore, the lack of warming of the cold events may be partly due to decadal changes in large-scale circulation

patterns. Several studies have demonstrated an increasing occurrence of the so-called warm Arctic – cold continents pattern in winter (Overland et al., 2011). Whether the pattern is due to sea ice loss and amplified warming in the Arctic (Cohen et al., 2013; Overland et al., 2021), SST-driven teleconnections from the North Pacific and North Atlantic (Yu et al., 2020), or inherent variability of the climate system (Blackport and Screen, 2020), in any case the pattern has partly compensated for the greenhouse warming over northern continental mid-latitudes in winter. Comparing the two halves of our study period, the

November-March warming at the 850 hPa level (based on ERA5) has two prominent spatial patterns: (1) the Arctic has warmed faster than Europe, and (2) eastern Europe has warmed faster than western Europe. However, the zonal difference in the warming rate is smaller than the meridional difference. Hence, an increasing portion of eastern origin of air masses responsible for cold events in Europe and a decreasing portion of their Arctic origin may have contributed to the lack of warming of extreme cold events in Europe, supposing that air masses of both origins have experienced enhanced diabatic cooling along

their trajectories.



The results we obtained were based on a novel methodology (Supplementary Fig. 1), of which we highlight and summarize the following aspects. First, the SOM method allows objective detection of the main features in pressure/geopotential and temperature patterns, without predefining their location, size, or shape. An obvious strength of the SOM method is its flexibility in 2-dimensional space compared to circulation indices and blocking detection, which primarily describe variability over a predetermined region. NAO and AO mostly describe variability in north-south direction, whereas GBI and UH in west-east direction. The SOM method can be regarded as an extension of those indices, as demonstrated in Supplementary Fig. 6. Previously, SOMs have, for example, been used for blocking detection to study European heat waves (Thomas et al., 2021). Second, we demonstrated a new way to utilize a combination of two separate SOM analyses and trajectory analyses. This captures dynamical features better than composites and allows cluster trajectories linked to certain types of cold anomalies. Third, the extensive trajectory analyses carried out demonstrated a large variability of processes causing the cold anomalies from one case to another but also within cases. This suggests that individual trajectories may not be representative for climatological aspects of the causes and origin of cold anomalies, and thus large ensembles are needed to yield statistically robust results on statistics about the relative importance of advection, diabatic and adiabatic processes, and their eventual changes in decadal time scales.

## 5 Conclusions

The SOM method, based on machine learning, enabled us to identify six principal cold anomaly types over Europe in winter. Importantly, no geographical regions for the temperature anomalies within Europe were predefined, which allowed us to capture their actual geographical location, extent, and shape. The goal was to find out where cold wintertime air masses come from and how the cold anomaly forms by investigating associated atmospheric large-scale circulation patterns combined with kinematic backward trajectories.

The location of the cold anomaly region was closely tied to the location of blocks; if the block is located farther in the east, also the cold anomaly is displaced eastwards. Our trajectory analyses, based on nearly two million individual trajectories, revealed that there are simultaneous warming and cooling (adiabatic and diabatic) processes, the net effect of which determines the final temperature anomaly. The air parcels are typically initially (5–10 d before) colder than at their arrival, but also initially warmer air parcels sometimes lead to cold anomalies over Europe. Most commonly the effect of adiabatic warming on the temperature anomalies is overcompensated by advection from regions that are climatologically colder than the target region, supported by diabatic cooling along the pathway. However, we found large regional differences in the contributions of different processes. Cold anomalies over western Europe and southeastern Europe are dominantly caused by advection, and over eastern Europe due to both advective and diabatic processes. The climate indices, such as the NAO index and GBI, provide a rough overview of the dynamical conditions linked to cold anomalies. The SOM analyses combined with extensive trajectory calculations that allow for the quantification of the three key processes causing cold anomalies provided a remarkable added value to describe and understand the dynamic and thermodynamic conditions leading to cold extremes over Europe.

We demonstrated that the extremeness of cold air masses has not decreased at the same rate with the warming climate, although
the Arctic source areas of cold air masses have experienced Arctic amplification. Hence, we also expect that the future response
of cold anomalies to climate warming will not solely depend on the increase of temperatures in the source areas, but may be
either partly compensated or even amplified by changes in the physical processes along the air mass pathways, and further
complicated by changes in the pathways themselves due to eventual changes in atmospheric circulation. Understanding the
mechanisms causing future changes in cold air masses will require in-depth analyses on both large-scale circulation and the
physical (adiabatic and diabatic) processes.

**Code availability**

LAGRANTO can be downloaded freely from https://iacweb.ethz.ch/staff/sprenger/lagranto/download.html. Code for the
analysis is available upon request.

**Data availability**

ERA5 data are freely available from the Copernicus Climate Data Store (https://cds.climate.copernicus.eu, last access: 30
May 2022; C3S, 2017; DOI of single level data: https://doi.org/10.24381/cds.adbb2d47; Hersbach et al., 2018). Daily NAO
and AO indices were obtained from the websites of Climate Prediction Center of National Oceanic and Atmospheric
Administration (NOAA) (https://www.cpc.ncep.noaa.gov/products/precip/CWlink/pna/nao.shtml,
https://ftp.cpc.ncep.noaa.gov/cwlinks/norm.daily.ao.index.b500101.current.ascii, last access: 30 May 2022). Daily values of
GBI were obtained from NOAA (https://psl.noaa.gov/gcos_wgsp/Timeseries/Data/gbi.day.data, last visit 25 October 2022).

**Acknowledgements**

The work of Tiina Nygård and Tuomas Naakka was supported by the Academy of Finland (AFEC project, contract 317999),
and the work of Timo Vihma by the European Commission's Horizon 2020 research and innovation framework program
under Grant agreement no. 388 101003590 (PolarRES project). We thank Michael Sprenger (ETH Zürich) for his support of
LAGRANTO.

**Author contributions**

TiN designed the study in collaboration with LP, TuN and TV. TiN carried out the analysis in collaboration with LP. All the
authors contributed to interpretation of the results and writing.

**Competing interests**

No competing interests.



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
