# Peer review of "Cold wintertime air masses over Europe: Where do they come from and how do they form?"

_EGUsphere, 2023_

## Author Comment (AC1)

*Dear Editor and Reviewers,*

*We highly appreciate all the constructive comments and suggestions we received in order to improve our manuscript and we have tried our best to take them into account in this revised manuscript. Our response is written in italics after each comment.*

**RC1**: 'Comment on egusphere-2023-889', Anonymous Referee #1

Summary of the work: The analysis considers cold air outbreaks from 1979-2020 in Europe using SOMs, to investigate the thermodynamic pathways for the extreme cold events. Despite the general warming, these extreme cold events have not weakened, so the authors partition the 1979-2020 period into two halves to investigate the changes to the thermodynamic contributions to cold events. The analysis uses kinematic back trajectories to investigate the formation of the cold air and makes some assumptions about the sources of advective, adiabatic, and diabatic changes in airmass temperature. The analysis shows that the general warming is compensated by diabatic cooling at decadal scales.

Review discussion: Overall, this analysis is solid, and the methods are sound. The analysis of the thermodynamic changes to parcel/airmass trajectories follows the methods of Röthlisberger and Papritz (2023), which is very appropriate for the goals of the research. However, the motivation for using separate SOM analyses for the interpretation of European cold events is not as well motivated/explained as perhaps it could be. Currently, it is fair but with more clarity and detail, it could be excellent. The SOM analysis for temperature/cold, makes sense, but then attributing the cold clusters to the individual nodes on a separate MSLP SOM that don't necessarily align with the notes in the temperature SOM is a bit confusing.

*We thank the reviewer for this comment about the methods. This study aims to provide a comprehensive picture of thermodynamic pathways of the formation of extreme cold air masses and their links to circulation. We acknowledge that our approach to apply two separate SOM analyses, one for 850 hPa temperature and the other one for MSLP, may cause confusion. In many of our previous studies, we have applied a single SOM analysis, but, in this manuscript, we see clear benefits of the advanced approach with two SOM analyses. For example, it reveals a wide variety of circulation patterns that can cause similar temperature anomalies, which is a novel insight that could not have been obtained without two separate SOM analyses. We are not aware of any other study with a similar approach with combination of two SOM analyses, and therefore we agree that additional explanation is required. Due to complexity of the methodology, the manuscript has a separate figure in the Supplements (Supplementary Fig. 1) to clarify how the different methods were combined.*

*As our primary goal is to study cold temperature anomalies, it was most straight-forward to cluster the 850-hPa temperature anomalies, instead of any other variable, using a SOM analysis. By doing so, we could group the most similar temperature anomaly fields together. To answer our research question, what kind of atmospheric circulation is linked to those cold anomalies, we had the following methodological options: (1) to investigate individual circulation fields during the cold anomaly cases separately, (2) to form composites of MSLP and 500 hPa geopotential fields for each cold anomaly type and (3) to make a separate SOM analysis for MSLP fields to cluster circulation types and connect those "generic" circulation types to cold anomaly types. Option (1) was considered too laborious, and most importantly, it would not have provided a clear, comprehensive and sufficiently condensed picture of the associated circulation types. Options (2) and (3) were included and compared in the study. The results of option (2) are shown in Figs. 3 and 4 (second columns) in a form of composites. Results of option (3) are*

*widely presented in the manuscript. Results based on composites (option 2) and MSLP SOM circulation types (option 3) are compared throughout the result Section 3.*

*The main motivation for this comparison is to see whether the direct MSLP and 500 hPa geopotential composites suffer from averaging of very different circulation types, and thus display an average circulation field, features of which do not occur as such. Our comparison in Section 3.2 indicates that option (2), relying only on the temperature SOM analysis, is not adequate for most of the cold anomaly types, especially for cold anomaly Types II and IV, due to the above-mentioned averaging problem. Hence, our combination of two separate SOM analyses has clear benefits.*

*We have now added this additional clarification to Section 2.2.1:*

*"Instead of only associating composites of MSLP fields to the results of temperature SOM, the separate SOM made of MSLP helped us to avoid averaging of very different pressure fields, which may result in misinterpretation of prevailing circulation features. Hence, this approach, with two separate SOMs, increased accuracy of our assessment of associated circulation patterns."*

*We have also added to Discussion the following:*

*"Second, we demonstrated a new way to utilize a combination of two separate SOM analyses and trajectory analyses. We report clear benefits of combining two SOM analyses, one for the main variable of interest (850 hPa temperature) and the other one representing atmospheric circulation (MSLP), as it reduces the risk for misinterpretation of averaged pressure fields. The SOM approach also allows to specifically cluster trajectories based on the cold anomaly type."*

Some of the confusion comes from the fact that these additional SOMs are only shown as supplemental figs. I see that you get more information about the flow configurations contributing to each type of cold type with the additional SOM analysis, but when the common MSLP or GPH modes represent less than 20% of the cold nodes/cases (e.g., type 1 in Fig. 3 & 4), is this the best way to present the result? Perhaps an analysis more like the chicklet plot in Supplemental Fig. 7 would be better suited for the discussion in the main text. Such a change would allow for broader explanation & description of the variety of flow configurations in each cold type than the current Fig. 3/4 discussion. If you take such an approach, the MSLP and GPH SOM figures would need to be moved from supplemental figures to full figures.

*We thank the reviewer for these important considerations, how to best present and visualize our results. In fact, in the preparatory phase of this manuscript, we had Supplementary Figs. 3, 4 and 7 as a part of the main manuscript, as the reviewer suggests. However, we realized that those figures, with a lot of information, are very heavy for a reader to digest. Therefore, we have chosen a simplified way to present our results in the main figures by selecting the two most common MSLP SOM patterns and have left the more detailed and comprehensive figures to the Supplementary material. By this choice, a reader has less responsibility for interpretation of the figures, and we bear our responsibility to interpret the most important circulation features linked to the cold anomalies and deliver them in the text and in a form of rather clear and simple figures.*

*It is true, as the reviewer mentions, that the most common circulation types presented in Figs. 3 and 4 represent only 18–35% of the cold anomaly cases. The text is, however, not restricted to the two most*

*common circulation patterns, but based on the main features of the full variety of SOM circulation types given in the Supplementary Figs. 3, 4 and 7. We have now modified the text to clarify this in the end of Section 3.1:*

*"We also summarize the main circulation features linked to the cold anomaly types. However, for more detailed results, Supplementary Fig. 7 can be used as a reference to make connections between the cold anomaly types (the left column in Fig. 3), individual circulation types (Supplementary Figs. 3 and 4) and climate indices (Supplementary Fig. 5)."*

With all that being said,
L194-195: Typo – The years for your latter and earlier periods need to be swapped.
*Corrected.*

L209-212: When you consider the events in which the SOM composite cold anomalies are less than the 10[th] percentile, how many of the days (or what fraction of days) in each Type/SOM group reach the cold threshold vs have the right pattern but don't meet the threshold?

*This information is given in the methods Section 2.2.3:*

*"Note that only cold anomalies with at least 10 cold anomaly grid points on the 0.25° x 0.25° lat-lon grid were considered in the trajectory analyses. The number of cases for which trajectories were computed and the number of trajectory cases are shown in Table 1."*

*Thus, the number of days not meeting the threshold can be seen in Table 1, as the difference between N and N(trajectory). The number (and fraction) of days not meeting the threshold are: 6 (3%) for Type I, 17 (4%) for Type II, 22 (10%) for Type III, 5 (2%) for Type IV, 18 (6%) for Type V and 4 (2%) for Type VI. We do not add these numbers to the manuscript, because we think that it is sufficient to show N and N(trajectory) in Table 1.*

L345: The assessment here about wind is based on the composite fields, are the weak gradients a result of the composite smoothing or are the gradients actually weak in most/all the cases?

*Here, we have meant that the gradients are actually weak in most cases, not only in the composite. To clarify this, we have now modified the text in Section 3.2.3:*

*"Both the commonly weak 500 hPa geopotential gradients and high MSLP enable the cold air mass to stay over central Europe without being rapidly advected elsewhere."*

L553: "The more efficient radiative cooling is most probably related to the generally higher radiative cooling rate of warmer and moister air." This statement seems important in this discussion and could be examined in more detail, "most probably" is a bit hand-wavy. In general, this paragraph has several locations where suggestive wording is used but could be

*We agree that we have widely used suggestive wording in this paragraph. The reason is that the explanations we have provided are mostly suggestions and our best guesses, and not specifically proven in our study. Their investigation is out of the scope of this study but relevant for future work.*

*We have slightly modified the text in this paragraph to be more precise, and less suggestive, where appropriate. In particular, we have added some theoretical explanation to our statements about enhanced radiative cooling:*

*"The more efficient radiative cooling is most probably related to the generally higher radiative cooling rate of warmer and moister air. This is due to the fact that radiative cooling depends on temperature and emissivity of air mass, which is mostly controlled by amount of water vapour and clouds, especially liquid clouds. Long-term increases in the amount of water vapour and cloud water have been reported in the regions of most common cold airmass pathways (Nygård et al., 2020); even modest changes in cloud water content have a large effect on radiative cooling."*

Fig. 8: The text on this figure is hard to reach, please make the text bigger or bolder. What do the dots on the trajectories represent. The caption says these are 10 d trajectories, but there are not 10 dots, there are 6 dots per trajectory. I assume these dots represent every 2 days, please clarify? On the topic of clarity, can the dots be colored as the line color, in (b) there are several overlapping lines/dots that make the plot cluttered and hard to interpret.

*We have now modified Fig. 8 according to the suggestions. We also added an explanation to the figure caption about the dots that show the evolution of time at a 2-day interval.*

Supplemental Fig. 7: The SOM numbers are very hard to read.

*We have now modified Supplemental Fig. 7 so that the numbers are more visible.*

**RC2**: 'Comment on egusphere-2023-889', Anonymous Referee #2

Review of the paper "Cold wintertime air masses over Europe: Where do they come from and how do they form?" By Nygård et al. This is an interesting paper that provides a very thorough analysis of cold air outbreaks in Europe. However, I do think that the paper is maybe too broad and all-encompassing, at the penalty of becoming slightly overwhelming and thus perhaps not as readable as it could have been.

I'll note first that I have read the comments of one reviewer who submitted their comments before me.

I don't have many comments on the scientific content, which seems sound. It's more a matter of some choices that you've made.

*We thank the reviewer for this overview and comments. It has been our intention to provide a comprehensive picture of thermodynamic pathways linked to extreme cold air masses in this manuscript. We think that the most unique and valuable aspect of this manuscript is that it really gathers several processes, interactions and excessive amounts of data together, rather than focuses on those individually. This makes it possible to form a comprehensive picture of how cold wintertime airmasses form. We acknowledge that a side-effect of this is that the manuscript has become rather long and, especially Section 3.2, with results for all the cold anomaly types individually, is a bit heavy for a reader.*

I agree with Reviewer #1 that it's not completely clear why you subselect such a small subset of the SOMs. Just by eyeballing, it certainly looks like e.g. map 41 is very cold in Scandinavia, but you write that you only select maps that satisfy the cold criteria in continental Europe or the British Isles. Why do you make this choice? Scandinavia is part of Europe… Furthermore, your subdivision of the maps into groups seems to be subjective. Is group VI really that different from group V? This needs more motivation.

*The reason why we have sub-selected a small subset of 850 hPa temperature anomaly SOM nodes is that in this study we only focus on the coldest anomalies. To minimize subjectiveness, we have used 10th percentile as a threshold to select the SOM nodes with the coldest anomalies over Europe. We want to emphasize that the absolute threshold temperature is thus regionally varying and takes into account different climate conditions in different parts of Europe. The reviewer questioned why the SOM node 41 is not taken into further analyses, even if it seems to have a clear cold anomaly over Scandinavia. The reason for not selecting node 41 is that its anomaly does not fulfill the selection criterion of having temperatures below the 10th percentile.*

*We would also like to clarify that in this study, Scandinavia is part of continental Europe. We have followed the most common geographical definition of continental Europe, which includes the Scandinavian peninsula. We have modified a sentence in Section 2.2.2 accordingly:*

*"Those 14 SOM nodes (marked in Fig. 2), with the composite 850 hPa temperature anomaly colder than the percentile threshold (Fig. 1) somewhere over continental Europe (including the Scandinavian Peninsula) or the British Isles, were selected."*

*We agree that our subdivision of the nodes into groups is subjective, because it was difficult to define any meaningful quantitative measure to form the groups. The subdivision is based on the location of the region with temperatures below the 10th percentile and additional assessment of the extent of the cold anomaly region. Cold anomaly Types V and VI have a lot in common, but the main difference is that cold anomaly Type VI extends further to the west and covers most parts of Europe, whereas Type V is limited more to the east. Physical processes behind them are largely similar, but the differences are found in atmospheric circulation related to them.*

*We have now modified the text in Section 3.1 and written:*

*"The types were formed based on the location of the temperature anomaly colder than the 10th percentile and the extent of the entire cold anomaly region, which were subjectively assessed."*

*We have also added the following to Section 3.2.6 describing cold anomaly Type VI:*

*"Type VI consists of nodes 55 and 56, in which 850 hPa temperature is below the 10th percentile over the British Isles, southern Scandinavia and Baltic countries; the whole cold anomaly extends over most of Europe, except the southeastern corner (Figs. 2 and 3). Hence, the cold anomaly region of Type VI extends further west than in Type V."*

*The similarities of Types V and VI are addressed in several places in the discussion Section.*

When it comes to the description of the Cold anomaly types in Section 3.2, it's long, and quite frankly a bit boring to read. I think the reason for this is that you seem to follow a template for each of the groups. This makes it rigorous and useful as a reference, but it does feel quite repetitive.

I think you could also reduce the number of panels. In Fig. 3 for example, it's not clear why you would show three MSLP panels for each group. Some of them are very similar. It's also not clear, if one does not wish to read the SI, how the selection of the most common and second most common groups was done. Maybe this could be motivated more clearly, or maybe it would be better to reduce the number of panels.

Perhaps you could try to reduce the length of Section 3 substantially by, instead of going through a fixed template for each group, try to extract the most interesting features, i.e. the ones that stand out? The Discussion is an attempt at that.

This is meant as well-intentioned advice. I think the topic and the approach are interesting, but I think the paper will be more widely read if you drastically reduce the number of figures and text and focus more on the results that stand out.

*To clarify the purpose of three panels of MSLP in Fig. 3, we want to point out that the second column shows the direct composite of MSLP occurring simultaneously with a particular cold anomaly type. This composite does not rely on a separate SOM analysis for MSLP, but can be made based on the 850 hPa temperature SOM. The third and fourth columns show the two most common MSLP SOM nodes occurring during a particular cold anomaly type. Hence, these two columns are based on the separate SOM analysis made for MSLP, which is one of the unique methodological features of this study. Selection of the most common circulation types is directly based on their statistical occurrences, which are shown in Supplementary Fig. 7. We have now modified the captions of Figs. 3 and 4:*

*"Fig. 3 Composites of 850 hPa temperature in the six cold anomaly types (first column on the left), direct composites of MSLP in each of those cold anomaly types (second column), the most common MSLP circulation type (based on a separate circulation SOM) during each cold anomaly type (third column), and the second most common circulation type (based on a separate circulation SOM) during each cold anomaly type (fourth column). Each row shows results for a certain cold anomaly type. The numbers in the third and fourth columns indicate the MSLP SOM node number (see Supplementary Figure 3) and the percentage of cases associated with this circulation type in each cold anomaly type. "*

*"Fig. 4 Composites of 850 hPa temperature in the six cold anomaly types (first column on the left), direct composites of 500 hPa geopotential in each of those cold anomaly types (second column), the most common 500 hPa geopotential pattern (based on a separate circulation SOM) during each cold anomaly type (third column) and the second most common 500 hPa geopotential pattern (based on a separate circulation SOM) during each cold anomaly type (fourth column). Each row shows results for a certain cold anomaly type. The numbers in the third and fourth columns indicate the circulation SOM node number (see Supplementary Figure 3) and the percentage of cases associated with this circulation type in each cold anomaly type."*

*The two most common circulation types presented in Fig. 3 (and Fig. 4) represent only 18–35% of the cold anomaly cases. The text is, however, not restricted to the two most common circulation patterns, but based on the main features of the full variety of SOM circulation types given in the Supplementary Figs. 3, 4 and 7. (Please also see our response to Reviewer 1 about the methods.)*

*Motivation for comparison of MSLP composites (second column) and the most common MSLP SOM nodes (columns 3 and 4) is to see whether the direct MSLP composites suffer from averaging of very different circulation types, and thus display an average circulation field with features that do not occur as such. This comparison in Section 3.2 indicates that the direct composites, relying only on the temperature SOM analysis, are not adequate for most of the cold anomaly types, especially for cold anomaly Types II and IV, due to the above-mentioned averaging problem. Hence, we want to show all these MSLP panels to demonstrate that a separate SOM analysis for MSLP, in addition to the SOM for the variable of interest, provides a clearly advanced and more accurate view on the atmospheric circulation linked to cold anomalies over Europe.*

*We agree that a more compact paper would probably reach a wider group of readers. However, we think that for this topic, a rigorous paper, which can also be used as a reference, will better serve the scientific advance in the field. Importantly, the current format enables (a) the results to be separately found for a particular region within Europe and (b) a fair comparison of anomaly types.*